# Impact of pausing elective hip and knee replacement surgery during winter 2017 on subsequent service provision at a major NHS Trust: a descriptive observational study using interrupted time series

Tim Jones [1,2,3] Chris Penfold [1,2,3] Maria Theresa Redaniel,[1,3] Emily Eyles [1,3] Tim Keen,[4] Andrew Elliott,[4] Ashley W Blom [5] Andrew Judge [2]

For numbered affiliations see end of article.

**Correspondence to**
Dr Tim Jones;
Timothy.Jones@bristol.ac.uk

## ABSTRACT

**Objectives** To explore the impact of a temporary cancellation of elective surgery in winter 2017 on trends in primary hip and knee replacement at a major National Health Service (NHS) Trust, and whether lessons can be learnt about efficient surgery provision.

**Design and setting** Observational descriptive study using interrupted time series analysis of hospital records to explore trends in primary hip and knee replacement surgery at a major NHS Trust, as well as patient characteristics, 2016–2019.

**Intervention** A temporary cancellation of elective services for 2 months in winter 2017.

**Outcomes** NHS-funded hospital admissions for primary hip or knee replacement, length of stay and bed occupancy. Additionally, we explored the ratio of elective to emergency admissions at the Trust as a measure of elective capacity, and the ratio of public to private provision of NHS-funded hip and knee surgery.

**Results** After winter 2017, there was a sustained reduction in the number of knee replacements, a decrease in the proportion of most deprived people having knee replacements and an increase in average age for knee replacement and comorbidity for both types of surgery. The ratio of public to private provision dropped after winter 2017, and elective capacity generally has reduced over time. There was clear seasonality in provision of elective surgery, with less complex patients admitted during winter.

**Conclusions** Declining elective capacity and seasonality has a marked effect on the provision of joint replacement, despite efficiency improvements in hospital treatment. The Trust has outsourced less complex patients to independent providers, and/or treated them during winter when capacity is most limited. There is a need to explore whether these are strategies that could be used explicitly to maximise the use of limited elective capacity, provide benefit to patients and value for money for taxpayers.

## STRENGTHS AND LIMITATIONS OF THIS STUDY

⇒ Trends analyses using data obtained from the electronic health records of a local hospital National Health Service (NHS) Trust are informative for clinicians and service managers in monitoring changes in planning and delivery of elective surgery and could be regularly updated in near real time for monitoring.

⇒ The inclusion of wider hospital admissions data beyond the NHS Trust allows us to estimate the proportion of people within the Trust catchment area having NHS-funded treatment at independent providers.

⇒ We report the experience of one NHS Trust that is one of the larger elective orthopaedic centres—the findings may not be generalisable to or reflect the experience of other trusts.

⇒ Our study does not include privately funded, privately provided hip and knee surgery, which may also have been changing over time.

## INTRODUCTION

Primary hip and knee replacement operations are common planned elective surgical procedures. They are highly clinically effective for improving symptoms of pain and functional limitations and have been shown to be safe and cost-effective.[1–4] Around 100 000 hip[4] and over 100 000 knee operations[3] are carried out each year in the UK. Demand for these operations has been increasing substantially in recent decades[5] with an ageing population, rising levels of obesity and widening indications for surgery in younger patient groups.[3 4]

Orthopaedic services have become more efficient over time, with the length of hospital stay for primary hip and knee replacements

reducing from around 15 days in 1997 to roughly 5.5 days in 2014.[6] This is largely due to the introduction of 'fast-track' surgery and enhanced recovery services,[7] which reduce length of stay while maintaining patient safety and outcomes of surgery.[6] However, over the past decade, there has also been a reduction in the numbers of hospital beds and operating theatres available for hip and knee replacement patients.[8] Waiting lists for orthopaedic procedures have been growing over time, and the average time people wait for treatment once on the waiting list has also increased.[9]

Pressures on elective surgery are exacerbated during winter, when resources for planned surgery are often displaced by more acute, unplanned hospital admissions.[8] At the end of 2017, this led to all planned elective hip and knee replacement operations in England being cancelled for the whole of January.[10] Even before the COVID-19 pandemic, over half a million people were already on the waiting list.[11] Patients are having to wait longer with deteriorating severe pain and functional limitation, affecting their health and quality of life. The COVID-19 pandemic has had an even greater impact on cancelling planned elective surgery, with over 635 000 people waiting for hip and knee replacements in April 2021, more than 10% of these waiting over a year and over a third waiting longer than the 18-week target.[11]

The winter of 2017 provides a form of 'natural experiment', where elective capacity was intentionally reduced close to zero. A natural experimental design is a valid methodological approach to evaluate the impact of a range of events, policies and interventions, which are not under the control of researchers.[12] Researchers can use the variation in exposure that natural experiments generate to analyse their impact on health outcomes. This provides a form of quasiexperimental study, where we can explore trends in provision of elective surgery before and after winter 2017, which is a robust approach to explore real-world impact when randomisation is not possible.[13 14]

Our aim was to understand what happens after common, planned elective surgery is temporarily cancelled, and how this might inform optimum planning of elective surgery when capacity is limited, such as following the COVID-19 pandemic. We used interrupted time series (ITS) analysis to model trends in elective hip and knee replacement surgery for a major National Health Service (NHS) Trust from 2016 to 2019 and see how these were impacted by the withdrawal of elective surgery in winter 2017. We explored these trends by patient factors (age, sex, deprivation, number of comorbidities) and seasonality to see when demand was highest for different patient groups.

## METHODS

This study is a longitudinal observational descriptive study using routinely collected administrative information about patients admitted to a major NHS Trust for elective hip and knee replacements, 2016–2019. It was developed and reported according to the Reporting of studies Conducted using Observational Routinely-collected Data (RECORD) extension[15] to Strengthening the Reporting of Observational Studies in Epidemiology (STROBE) guidelines for observational studies using routinely collected data.

### Data sources

We used two data sources for our analyses. The first was an extract of elective primary hip and knee replacement inpatient admissions identified from the Trust's electronic medical records (EMR) between 1 January 2016 and 31 December 2019. Up to 29 diagnoses were provided per entry using the International Classification of Diseases version 10 (ICD-10), and up to 11 procedures were provided per entry using the Office of Population Censuses and Surveys Classification of Interventions and Procedures version 4 (OPCS-4). The extract included patient demographics such as age, sex, deprivation quintile and comorbidities; and other characteristics of the hospital admissions such as length of stay. This data source was used for all analyses of hip and knee replacements at the Trust, including those relating to patient demographics, length of stay and bed occupancy.

The second data source was pseudonymised national admitted patient care Hospital Episode Statistics (HES-APC) between 1 January 2016 and 31 December 2019. HES-APC is a routinely collected dataset that records all episodes of admitted (day case or inpatient) care provided to patients at NHS hospitals in England and to NHS-funded patients treated in independent hospitals.[16] Each episode represents a period of care under one consultant team. Up to 20 diagnoses and 24 clinical procedures are recorded per episode using ICD-10 codes and OPCS-4 codes, respectively. HES also includes the Lower Super Output Area (an area of around 1500 people) of residence for each patient, which can be linked to clinical commissioning group (CCG) of residence. This data source was used to estimate elective capacity overall at the Trust, and the ratio of public/private provision of hip and knee replacements in the catchment area for the Trust (see details below), which could not be gathered from the extract provided from the Trust EMR.

### Hospital admissions for hip and knee replacements

Hospital admissions for elective hip and knee replacements were identified by entries with a primary procedure code representing primary hip or knee replacement (online supplemental table T1) using the Trust EMR. We used this information to explore summary characteristics of the hospital admissions over time (overall counts of admissions, average age, proportion of women, proportion with 2+ comorbidities, proportion in the two most deprived quintiles) stratified by primary hip or knee replacements.

## Length of stay and bed occupancy

We used the average number of overnight stays in hospital (days) for length of stay, trimmed at 30 days to exclude a small number of outliers (n=32, 0.6%). Trimming allowed us to model averages assuming a roughly normal distribution, which we felt was more easily interpretable. Bed occupancy was the total number of beds used overnight for hip and knee replacement patients.

## Comorbidity of admissions

For each admission, we counted the number of conditions from the Charlson Comorbidity Index[17] recorded in the diagnosis fields. The Charlson index provides a summary of weighted scores relating to different comorbidities and has been shown to be associated with mortality. Admissions were categorised into those with zero, one, and two or more Charlson comorbidities.

## Ratio of elective to emergency admissions

To estimate the ratio of elective to emergency admissions for all purposes at the Trust (as a proxy for elective capacity), we extracted all hospital admissions from HES-APC with the Trust as a provider and categorised them into elective and emergency (admission method beginning with '1' or '2', respectively).

## Ratio of public to private provision of hip and knee replacements

To estimate the ratio of public to private provision of NHS-funded elective hip and knee surgery for the Trust catchment area, we extracted all hospital admissions for primary hip and knee replacements (codes in online supplemental table T1) for residents of the major local CCGs from HES-APC (using 2021 CCG boundaries after local CCGs had merged into one CCG[18]) and categorised providers into public and private (provider code beginning with 'R' or 'N', respectively).

## Statistical analysis

We explored the change in trend for the following outcomes before/after the winter 2017 cancellation of elective surgery, stratified by primary hip and knee replacements: number of hospital admissions; average age of patients; proportion of women; proportion with 2+ comorbidities; proportion in more deprived deprivation quintiles (4 and 5); average length of stay; bed occupancy; and ratio of public to private provision of surgery. Additionally, we explored the overall ratio of elective to emergency admissions at the hospital for any purpose without stratification. For each of the outcomes, we conducted ITS analyses using segmented regression models comparing hospital admissions in the 'before' period (January 2016 to November 2017) to the 'after' period (February 2018 to December 2019). We excluded the winter 2017 period when admissions were very low (December 2017 and January 2018). The ITS

**Table 1** Interrupted time series model results

| | Pre trend | | Level change | | Trend change | |
|---|---|---|---|---|---|---|
| | Estimate (95% CI) | P value | Estimate (95% CI) | P value | Estimate (95% CI) | P value |
| Hip, admissions | 1 (0.99 to 1) | 0.300 | 1.06 (0.91 to 1.22) | 0.469 | 0.99 (0.98 to 1.01) | 0.239 |
| Hip, age* | −0.01 (−0.1 to 0.07) | 0.737 | 1.57 (−0.1 to 3.24) | 0.065 | −0.06 (−0.17 to 0.05) | 0.307 |
| Hip, proportion of women | 1 (0.99 to 1) | 0.582 | 0.97 (0.86 to 1.08) | 0.549 | 1.01 (1 to 1.02) | 0.089 |
| Hip, Charlson | 0.99 (0.98 to 1.01) | 0.380 | 1.41 (1.06 to 1.87) | 0.017 | 1.01 (0.99 to 1.03) | 0.220 |
| Hip, deprivation | 1 (0.99 to 1.01) | 0.587 | 1.03 (0.87 to 1.21) | 0.754 | 1 (0.99 to 1.02) | 0.660 |
| Hip, length of stay (LoS)* | −0.01 (−0.03 to 0.02) | 0.660 | 0.31 (−0.19 to 0.82) | 0.225 | −0.01 (−0.05 to 0.02) | 0.425 |
| Hip, bed occupancy | 1 (0.99 to 1.01) | 0.643 | 1 (0.84 to 1.19) | 0.997 | 0.99 (0.97 to 1) | 0.149 |
| Hip, public–private* | 0.01 (−0.02 to 0.04) | 0.377 | −0.74 (−1.24 to 0.25) | 0.003 | −0.02 (−0.05 to 0.01) | 0.218 |
| Knee, admissions | 1 (0.99 to 1) | 0.106 | 0.84 (0.73 to 0.98) | 0.022 | 1 (1 to 1.01) | 0.256 |
| Knee, age* | −0.08 (−0.16 to 0) | 0.054 | −1.63 (−2.99 to 0.28) | 0.018 | 0.21 (0.12 to 0.31) | 0.000 |
| Knee, proportion of women | 1 (1 to 1.01) | 0.150 | 0.96 (0.85 to 1.08) | 0.513 | 0.99 (0.99 to 1) | 0.193 |
| Knee, Charlson | 1.01 (0.99 to 1.03) | 0.249 | 0.64 (0.46 to 0.89) | 0.009 | 1.04 (1.02 to 1.07) | 0.001 |
| Knee, deprivation | 1.01 (1 to 1.01) | 0.189 | 0.97 (0.79 to 1.19) | 0.758 | 0.99 (0.97 to 1) | 0.021 |
| Knee, LoS* | −0.02 (−0.05 to 0) | 0.058 | 0.18 (−0.28 to 0.63) | 0.449 | −0.01 (−0.04 to 0.02) | 0.566 |
| Knee, bed occupancy | 0.99 (0.98 to 1) | 0.103 | 0.83 (0.7 to 0.99) | 0.037 | 1 (0.99 to 1.01) | 0.993 |
| Knee, public–private* | 0.01 (−0.02 to 0.03) | 0.667 | −0.48 (−1.03 to 0.07) | 0.090 | −0.02 (−0.04 to 0.01) | 0.225 |
| Elective to emergency ratio* | −0.01 (−0.01 to 0) | 0.171 | −0.32 (−0.45 to 0.2) | 0.000 | −0.02 (−0.03 to 0.01) | 0.003 |

*Linear regression model (additive) rather than Poisson regression model (multiplicative).

analyses explored the 'pre trend' before winter 2017, and how this trend changed after winter 2017,[12 19] allowing for an immediate 'level change' up or down in February 2018, and a longer-term 'trend change' in the slope afterwards. We explored seasonality in the data by including indicator variables for spring, summer and autumn[19] compared with winter as a baseline, and adjusted for serial autocorrelation using Newey-West standard errors with a maximum lag of two.[20–22] For count or proportion outcomes (number of admissions, proportion women, proportion with 2+ comorbidities, proportion in top two deprivation quintiles, bed occupancy), segmented Poisson regression models were fit to the data, while for averages/ratios (average age, average length of stay, ratio of elective to emergency admissions, ratio of public to private provision), segmented linear regression models were fit, using the 'glm' command in Stata. Sensitivity analyses were conducted adjusting the maximum lag for serial autocorrelation to zero and five; this would not affect point estimates but could alter standard errors, confidence intervals and p values.

All statistical analyses were conducted using Stata/MP V.16.1. Smoothed trends were fit to the data on all plots using the 'lowess' command with bandwidth 0.3. Stata code is available at: https://github.com/jonestim2002/hdr_uk_hospital_efficiency, which is publicly accessible.

### Patient and public involvement

Initial research ideas for the grant application of which this work is part were presented to the public in a workshop and suggestions and comments were incorporated in the protocol. Feedback during the workshop was positive, with participants agreeing with the research objectives and the identified need.

## RESULTS

### Descriptive information and demographics

A total of 2623 patients had a hip replacement and 2674 had a knee replacement at the Trust in the 4 years between 2016 and 2019. The mean age of patients was 67 years and 60% were women for both types of operations.

### Trend changes after winter 2017

Table 1 shows the results of our ITS analyses for all outcomes, including the trend before winter 2017 (pre trend), any immediate change after winter 2017 (level change) and any change in the slope after winter 2017 (trend change). These are described in more detail below.

### Trends in hip and knee elective hospital admissions over time

The overall numbers of elective primary hip and knee replacement operations gradually declined over the study period, from 63 hip and 65 knee replacements per month in 2016 to 49 hip and 51 knee replacements per month in 2019. While there was a drop off in winter 2017, after elective surgery was restarted hip replacements resumed at similar numbers and continued to decline along a similar trajectory. Numbers of knee replacements dropped by 16% after winter 2017 (level change=0.843, 95% CI: 0.728 to 0.976, p=0.022) and the slope appeared to level off, although there was little evidence for this in the regression model (trend change=1.005, 95% CI: 0.996 to 1.014, p=0.256; see figure 1 and online supplemental table T2).

### Age on admission

There was a change in the trend in average age for knee replacements after winter 2017 (trend change=+0.21, 95% CI: 0.12 to 0.31, p<0.001) towards treating older patients over time (+1.59 years of age per year; see figure 2).

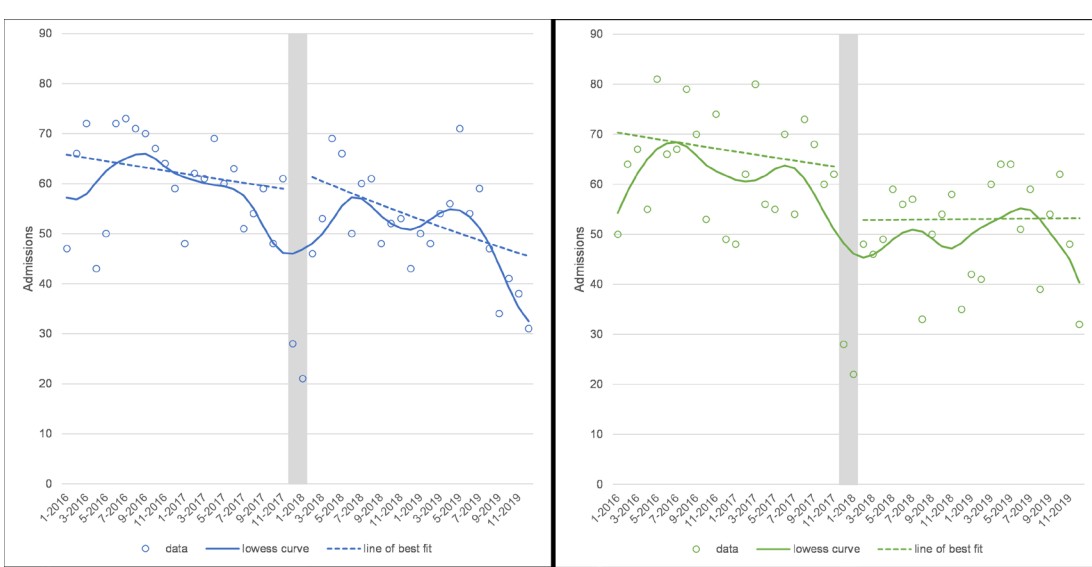

**Figure 1** Elective hip (left panel) and knee (right panel) replacement admissions at the Trust. Note: grey area shows the winter 2017 cancellations and is excluded from the analysis.

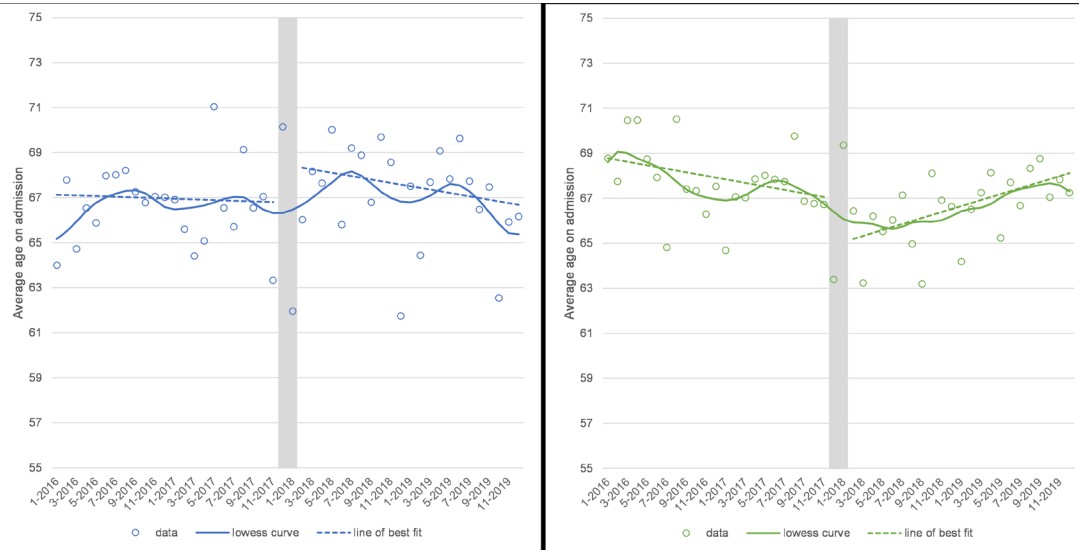

**Figure 2** Average age on admission for hip (left panel) and knee (right panel) replacements at the Trust. Note: grey area shows the winter 2017 cancellations and is excluded from the analysis.

## Comorbidity of admissions

There was a level change upwards in the proportion having hip replacements with 2+ comorbidities after winter 2017 (level change=1.411, 95% CI: 1.064 to 1.873, p=0.017) and an upward slope change for knee replacements (trend change=1.042, 95% CI: 1.017 to 1.067, p=0.001; see figure 3).

## Deprivation

There was evidence of a reducing proportion of the most deprived people having knee replacements after winter 2017 (trend change=0.986, 95% CI: 0.974 to 0.998, p=0.021).

## Ratio of elective admissions to emergency admissions at the Trust

There was an overall downward trend in the ratio of elective to emergency admissions at the Trust, from an average of 2.91 (SD: 0.17) electives for every emergency in 2016 to 2.16 (SD: 0.06) in 2019 (see online supplemental figure F1). The ratio reduced after winter 2017 (level change=−0.322, 95% CI: −0.446 to −0.198, p<0.001) and started to decrease more rapidly afterwards (trend change=−0.016, 95% CI: −0.026 to −0.005, p=0.003).

## Ratio of public to private provision of hip/knee elective surgery at the Trust

There was evidence of a level change downwards in public provision compared with private provision after winter 2017 for both types of surgery, but particularly for hip replacements (hips level change=−0.741, 95% CI: −1.237

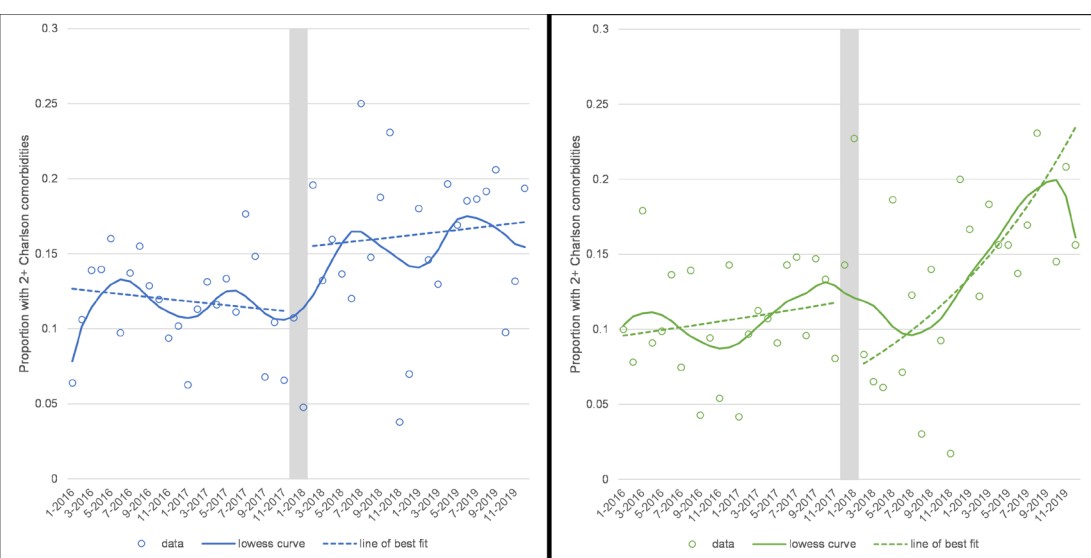

**Figure 3** Proportion of people having hip (left panel) and knee (right panel) replacements with 2+ Charlson comorbidities recorded. Note: grey area shows the winter 2017 cancellations and is excluded from the analysis.

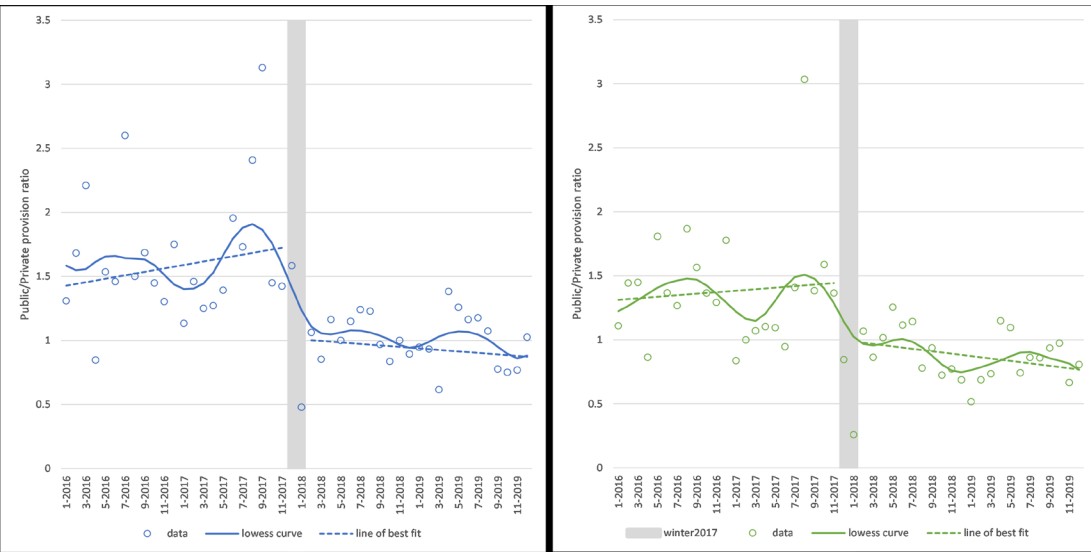

**Figure 4** Ratio of public to private provision of elective hip (left panel) and knee (right panel) replacements for National Health Service patients in the Trust clinical commissioning group. Note: grey area shows the winter 2017 cancellations and is excluded from the analysis.

to −0.245, p=0.003; knees level change=−0.476, 95% CI: −1.026 to +0.074, p=0.09; see figure 4).

### Bed occupancy

For hip and knee replacements, bed occupancy has reduced over time, although there was not evidence of this in the regression model for hip replacements, and there was a level change downwards (level change=0.834, 95% CI: 0.704 to 0.989, p=0.037) for knee surgery after winter 2017 (see figure 5).

### Length of stay

The average length of hospital stay was 5.5 days (SD: 5.9 days) for hip replacements and 5.2 days (SD: 5.0 days) for knee replacements in 2016, compared with 5.1 days (SD: 4.1 days) and 4.3 days (SD: 3.4 days), respectively, in

2019 (see online supplemental figure F2). However, there was no evidence in the regression models for a change after winter 2017.

### Seasonality

Online supplemental table T2 shows seasonality results for each of our ITS analyses. Hip and knee operations were clearly seasonal, with higher admissions in non-winter months compared with winter; 21% higher in the highest season (summer) for hips (summer=1.207, 95% CI: 1.094 to 1.332, p<0.001) and 31% higher in the highest season (spring) for knee replacements (spring=1.308, 95% CI: 1.157 to 1.479, p<0.001), excluding winter 2017. Bed occupancy for both types of operation was also seasonal, with lower occupancy in the winter months compared

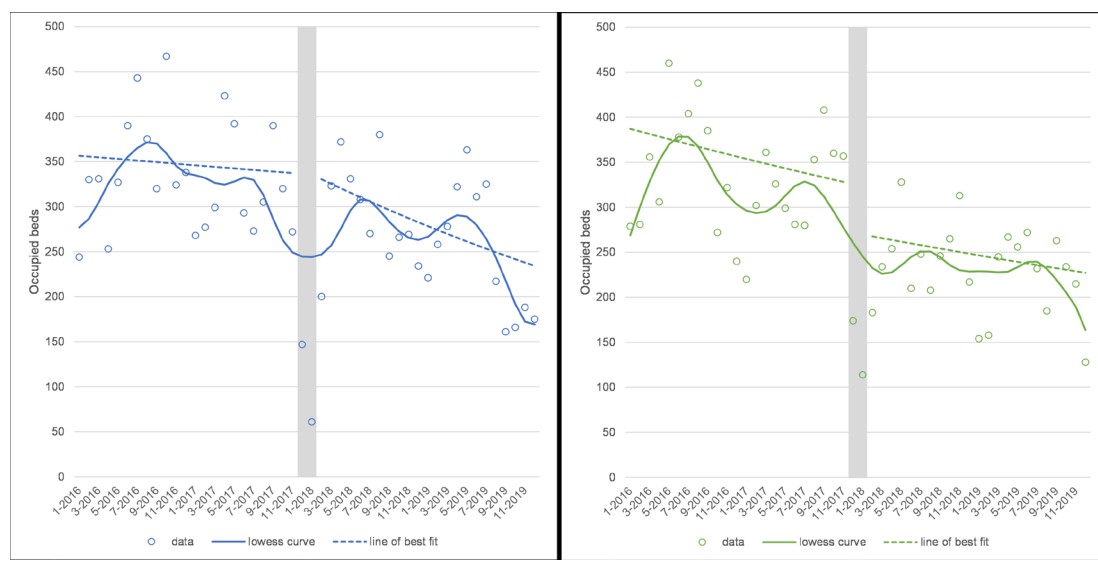

**Figure 5** Bed occupancy for hip (left panel) and knee (right panel) replacements at the Trust. Note: grey area shows the winter 2017 cancellations and is excluded from the analysis.

with all other seasons (see online supplemental table T2); for example, summer bed occupancy was 324 beds for hips and 291 beds for knees on average compared with winter bed occupancy of 225 beds for hips and 199 beds for knees on average. Length of stay was longer in spring than winter for hip replacements (spring=+0.502 days, 95% CI: 0.214 to 0.79, p=0.001), and longer in spring (+0.422 days, 95% CI: 0.073 to 0.771, p=0.018) and autumn (+0.396 days, 95% CI: 0.015 to 0.777, p=0.042) compared with winter for knee replacements.

The ratio of public to private provision was higher in the summer (1.56 for hips and 1.28 for knees) compared with winter (1.22 and 0.99, respectively) months (hips summer=+0.308, 95% CI 0.154 to 0.463, p<0.001; knees summer=+0.276, 95% CI: 0.035 to 0.517, p=0.025).

There was also some evidence of seasonality in the types of patients being admitted for hip and knee replacements. For hip replacements, the mean age of patients was 66 years in winter compared with 68 years in summer (summer=+2.09; 95% CI: 0.81 to 3.37, p=0.001); a higher proportion was performed on women in the summer (64%) compared with winter (58%) months (summer=1.088, 95% CI: 1.001 to 1.183, p=0.048); and a higher proportion of people had 2+ comorbidities in the summer (15.9%) compared with winter (12.3%) months (summer=1.306, 95% CI: 1.096 to 1.557, p=0.003). For knee replacements, there was a higher proportion of more deprived people (quintiles 4 and 5) in the spring (37.6%) compared with the winter (30.2%) months (spring=1.224, 95% CI: 1.077 to 1.49, p=0.002).

## DISCUSSION
### Principal findings
The temporary cancellation of elective services during winter 2017 does appear to have had some impact on service provision at the Trust after that time. There was an immediate and sustained reduction in the number of knee replacements being done at the Trust and this was also reflected in the drop in bed occupancy for knee surgery. The average age for knee replacement and comorbidity of hip and knee surgery patients increased after winter 2017, while the proportion of more deprived people having knee replacements decreased, and the ratio of public to private provision of hip and knee replacements in the local area dropped after winter 2017. This suggests an NHS-funded outsourcing of less comorbid hip and knee replacement surgery to independent providers, and therefore on average, the patients being treated at the Trust became older and more comorbid. There was a general decrease in capacity for elective surgery at the Trust (ratio of elective to emergency admissions), mostly driven by increasing non-elective admissions even before the COVID-19 pandemic. The winter 2017 cancellation may have been just one symptom of this overall pressure on elective surgery that underlies some of the longer-term changes in provision.

There was also some seasonality in service provision. It is no surprise that elective admissions and bed occupancy are lower in winter when the hospital requires capacity for an increase in unplanned admissions. There were also indications that people being admitted in winter were younger, less comorbid and less deprived (particularly for knee surgery). Length of stay for hip and knee replacements was lower in winter compared with spring. This suggests the admission of younger, less comorbid patients during the winter months given the reduced elective capacity and delaying surgery for more comorbid patients to when capacity is higher in the following months.

### Strengths and limitations
Trends analyses such as these, using data obtained from the EHR of a local hospital NHS Trust, are informative for clinicians and service managers in monitoring changes in planning and delivery of elective surgery and could be regularly updated in near real time for monitoring. This concept might be informative for other commissioning groups/ Trusts to adopt for monitoring of their own elective surgery and capacity. We report the experience of just one trust that is one of the larger elective orthopaedic centres, and hence the findings may not be generalisable to or reflect the experience of other trusts. Our findings are observational and report changes observed at the Trust following cancellation of elective services in winter 2017; further work would be needed to understand the impact of any changes on outcomes such as throughput of patients, waiting times, waiting lists, outcomes of surgery, costs and equity of access to surgery. There is likely to be some correlation between the covariables explored; for example, older people tend to be less deprived and have more comorbidities, which may account for some of our results. Some of the increase in age at operation after winter 2017 may be due to increased waiting times for surgery. We should be aware that some results may reflect chance findings due to multiple testing and type 1 error. The trends in the data as plotted do not change substantially in sensitivity analyses accounting for different autocorrelation lags (online supplemental tables T3 and T4). The catchment area of the Trust is not exactly the same as the major local CCG and is difficult to define exactly. However, 89.4% of admissions at the Trust were for residents of the local CCG and we felt this was a reasonable approximation to estimate the ratio of public to private provision in the Trust catchment area. Our analyses only include NHS-funded surgery and not privately funded, privately provided surgery.

### Comparison to other studies
A previous study[23] using data for England from HES found increasing private provision of elective hip arthroplasties nationally from 2007/2008 to 2012/2013, particularly for less deprived people, which echoes our findings. More recent news stories have suggested that 20% of NHS-funded hip replacements and 29% of NHS-funded knee replacements were carried out by independent providers in 2016/2017[24] and that independently provided hip and knee replacement surgery (privately or NHS funded) has now overtaken NHS provision.[25] A UK-wide study[6] using primary care data linked to hospital admissions found

similar effects of patient characteristics (age, sex, comorbidity and deprivation) on length of stay for primary hip and knee replacements, although they did not explore seasonality. A recent qualitative study[26] highlighted the negative financial and emotional impact of winter elective cancellations on patients and their families and recommended better advanced planning of elective operations to reduce these impacts.

## Implications for clinicians and policy-makers

Outsourcing of less complex hip and knee replacements to take advantage of spare capacity in non-NHS hospitals may be a good strategy to reduce waiting times and waiting lists for surgery and get the best results for patients, given the evident capacity limitations. It is a strategy that has already been used in other NHS Trusts,[27] and outsourcing more generally is recognised by the British Medical Association as a short-term solution to reducing waiting lists, although they recommend this goes alongside a longer-term commitment to increased NHS capacity.[28] The evidence is unclear regarding the impact of private provision on quality of care for patients and value for money for the public sector,[29 30] with some studies indicating potentially lower quality of healthcare.[31] There are questions about how much it increases staff capacity because some staff members transfer from public to private practice.[28 29] It would also leave the NHS Trust to cope with more complex cases,[30] which could have a detrimental impact on their service.

There are training implications of outsourcing because trainee surgeons are usually trained in NHS hospitals by first undertaking less complex cases on healthier patients. Trainees can find they are redeployed away from training to cover for a lack of trained staff in the public sector, which may be detrimental to their training and potentially harmful for patients.[32 33] The Royal College of Surgeons offers guidance around appropriate redeployment of trainees.[34]

There are also potential equity implications of outsourcing, if less complex cases have the option of surgery with shorter waiting times at independent providers, while more complex (and potentially more deprived) cases do not. We would need to consider the acceptability of this outsourcing to patients and practitioners, and the quality of patient outcomes.

There is an indication that some selection of patients for elective surgery depending on available capacity already takes place at the Trust. It is possible that this could become a more explicit strategy, based on evidence, to optimise the use of limited capacity in hospitals at different times of the year. However, this could mean that people placed earlier on the waiting list for surgery might get their surgery later due to such scheduling strategies, so acceptability to patients would need to be explored. We need to understand how the scheduling and possible outsourcing of elective surgery for different types of patients, depending on capacity, may impact on throughput of patients, waiting times, waiting lists,

outcomes of surgery, costs and equity of access to surgery. An appropriate balance would need to be achieved to maximise the benefits for patients, and research is needed to understand what that balance is. Additionally, we need to understand whether this type of scheduling and outsourcing is acceptable to people waiting for hip and knee surgery as well as clinicians. These issues of optimising limited elective resources are in even sharper focus due to the backlog in waiting lists caused by the COVID-19 pandemic.

## Unanswered questions and future research

We need to understand how the scheduling and possible outsourcing of elective surgery for different types of patients, depending on capacity, may impact on throughput of patients, waiting times, waiting lists, outcomes of surgery, costs and equity of access to surgery. Inevitably, outsourcing simpler patients to the independent sector will leave more complex patients being treated by NHS Trusts, which could have a detrimental impact on their service. An appropriate balance would need to be achieved to maximise the benefits for patients, and research is needed to understand what that balance is. Additionally, we need to understand whether this type of scheduling and outsourcing is acceptable to people waiting for hip and knee surgery as well as clinicians.

## CONCLUSIONS

Declining elective capacity and seasonality has a marked effect on the provision of joint replacement, despite efficiency improvements in hospital treatment. The Trust has outsourced less complex patients to independent providers and/or treated them during winter when capacity is most limited. There is a need to explore whether these are strategies that could be used explicitly to maximise the use of limited elective capacity, provide benefit to patients and value for money for taxpayers.

**Author affiliations**
[1]NIHR ARC West, University Hospitals Bristol and Weston NHS Foundation Trust, Bristol, UK
[2]Musculoskeletal Research Unit, University of Bristol, Bristol, UK
[3]Population Health Sciences, Bristol Medical School, University of Bristol, Bristol, UK
[4]North Bristol NHS Trust, Westbury on Trym, Bristol, UK
[5]Translational Health Sciences, Bristol Medical School, University of Bristol, Bristol, UK

**Acknowledgements** The study reported here would not have been possible without information provided by the NHS Trust, and the use of their hospital admissions data via the NIHR ARC West Partnership Agreement.

**Contributors** TJ contributed to study design, data cleaning, data analysis, interpretation of results and writing the manuscript. MTR contributed to study conceptualisation, supervision, interpretation of results and reviewing the manuscript. TK contributed to data curation, supervision, interpretation of results and reviewing the manuscript. AE contributed to data curation, interpretation of results and reviewing the manuscript. CP and EE contributed to interpretation of results and reviewing the manuscript. AWB contributed to study conceptualisation, supervision, interpretation of results and reviewing the manuscript. AJ contributed to study conceptualisation and design, supervision and writing the manuscript. TJ

had full access to the data in the study and takes responsibility for the integrity of the data and the accuracy of the data analysis.

**Funding** This research was funded by Health Data Research UK (HDR UK) Better Care South-West Partnership. EE, TJ and MTR's time was supported by the National Institute for Health Research Applied Research Collaboration West (NIHR ARC West). AJ and CP were supported by the NIHR Biomedical Research Centre at University Hospitals Bristol and Weston NHS Foundation Trust and the University of Bristol. The views expressed in this article are those of the author(s) and not necessarily those of the NHS, the NIHR, the Department of Health and Social Care or HDR UK.

**Competing interests** TJ, EE and MTR had financial support from NIHR ARC West for the submitted work. AJ has had financial support in the previous 3 years through institutional grants from NIHR, HDR UK, Versus Arthritis, Healthcare Quality Improvement Partnership (HQIP), Royal College of Physicians (RCP) and Health Foundation, had unpaid committee or leadership roles relating to musculoskeletal conditions for NIHR, Nuffield Foundation, Warwick CTU, and Versus Arthritis and a paid expert panel role for Nuffield Foundation Oliver Bird Fund; no other financial relationships with any organisations that might have an interest in the submitted work in the previous 3 years; no other relationships or activities that could appear to have influenced the submitted work.

**Patient and public involvement** Patients and/or the public were involved in the design, or conduct, or reporting, or dissemination plans of this research. Refer to the Methods section for further details.

**Patient consent for publication** Not applicable.

**Ethics approval** Not applicable.

**Provenance and peer review** Not commissioned; externally peer reviewed.

**Data availability statement** Data may be obtained from a third party and are not publicly available.

**ORCID iDs**
Tim Jones http://orcid.org/0000-0002-1199-8668
Chris Penfold http://orcid.org/0000-0001-8654-353X
Emily Eyles http://orcid.org/0000-0002-2695-7172
Ashley W Blom http://orcid.org/0000-0002-9940-1095
Andrew Judge http://orcid.org/0000-0003-3015-0432

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
