## [Reviewer comments · BMJ Open]

ARTICLE DETAILS

TITLE (PROVISIONAL)	The impact of pausing elective hip and knee replacement surgery during winter 2017 on subsequent service provision at a major NHS Trust: a descriptive observational study using interrupted time series
AUTHORS	Jones, Tim; Penfold, Chris; Redaniel, Maria Theresa; Eyles, Emily; Keen, Tim; Elliott, Andrew; Blom, AW; Judge, Andrew

VERSION 1 – REVIEW

REVIEWER	Kulkarni, Kunal University Hospitals of Leicester NHS Trust, Trauma & Orthopaedics
REVIEW RETURNED	29-Oct-2022

GENERAL COMMENTS	Thank you for asking me to review this paper. The authors should be commended for performing this large piece of work. I appreciate that analysis of Trust and HES data for multiple years takes much time and effort. The methodology is certainly robust. However, while the observations made are interesting, I do not believe the authors actually address their initial study aims. Overall, I unfortunately have two main concerns with the study, namely 1) the "so what" (i.e. how does this actually benefit readers in improving patient care) and 2) generalisability to other populations/regions. Some comments/thoughts for potential improvement are below: - The authors highlight that they have not included data from the independent sector; however, as they also comment, "the ratio of public to private provision of hip and knee replacements in the local area dropped after winter 2017". Given that the independent sector has played a role in most regions during recovery periods, it would be interesting to know what trends these (perhaps not nationally insignificant) numbers highlight. I appreciate that obtaining this data may not be possible, although as one of the goals is to explore trends, this does represent a data gap, particularly given the range of different independent providers in different regions.- The authors suggest that the merits/uses of these are through "monitoring changes in planning and delivery of elective surgery", although they also state that "the catchment area of the Trust is not exactly the same as the major local CCG and is difficult to define exactly". This raises the question of generalisability, both
--

	locally, and subsequently nationally to other regions where policies and strategies may have differed on working through waiting lists.  - In their conclusions, the authors highlight 2-3 existing studies that each agree with part of this study's conclusions, raising a query about the novelty of these findings and the unique points they bring to the evidence base. There are also other similar studies that I have read during the past couple of years that highlight similar trends observations peri-COVID. - With regard to suggestions/strategies/approaches going forward, the authors state "outsourcing of less complex hip and knee replacements to take advantage of spare capacity in non-NHS hospitals may be a good strategy to reduce waiting times and waiting lists for surgery and get the best results for patients given the evident capacity limitations - while I do not disagree with this statement (and indeed this has been policy many regions both, peri-pandemic and during past winter), this is not a statement specifically supported by data from the current study, and I do not believe this can therefore constitute a true 'learning point', particularly when no supplementary material is provided on how exactly patients could be prioritised for this approach (other than being low ASA). - Ultimately, the aim of this study was to "understand what happens after common, planned elective surgery is temporarily cancelled, and how this might inform optimum planning of elective surgery when capacity is limited, such as following the COVID-19 pandemic." However, the discussion unfortunately fails to highlight pragmatic and evidence-based solutions to the changes in capacity/throughput observed. The authors conclude "the Trust has outsourced less complex patients to independent providers, and/or treated them during winter when capacity is most limited. There is a need to explore whether these are strategies that could be used explicitly to maximise the use of limited elective capacity, provide benefit to patients, and value for money for taxpayers" - leaving this initial aim of 'informing optimum planning' unanswered. In summary, while the highlighted observations are interesting, my main concern is that this study in its current form unfortunately fails to clearly address what can be done to resolve the issues highlighted, and therefore I am unclear how readers with similar problems from other regions will find this of interest in tangibly improving patient care.
--	--

REVIEWER	Gray, William
	NHS Improvement, Getting It Right First Time
REVIEW RETURNED	02-Nov-2022

GENERAL COMMENTS	This is a paper on an important and interesting topic. The findings have direct relevance to the current interest in recovery of elective services following the COVID-19 pandemic and in building future resilience into elective surgery services. I have one overarching comment and a few smaller ones. Major comment  1. The methods and results sections are difficult to follow in places and it is not clear how the data in the text of the results section relate to the figures and tables. As an example (although this is consistent throughout the results section) the first mention of the regression models in the text is 'Numbers of knee replacements dropped by 16% after winter 2017 (RR=0.843, 95% CI: 0.728 to 0.976, p=0.022)', . It takes a good deal of searching to realise that
---

this is referring to the level change in Supplementary material T2 and only at this point was I able to understand what the figures were referring to. The fact that the abbreviation RR is not pre-defined does not help. I have a few suggestions that might help make the results section easier to follow, although you may not feel that all these need to be followed and there is no intention to be prescriptive.

1. A simple descriptive table (Table 1) of the patient characteristics at the pre and post periods would be helpful for orientation. As this is an online journal, I assume table/figure limits are not a major issue.
2. ITS analysis can be rather dense, and the figures are a vital visual summary, but I feel the pre-trend, level change and post-trend data in Table T2 would be better moved to the main manuscript as a new table 2 as it is central to the results being presented. The seasonality data is fine as supplemental.
3. The results text should link more closely to the data and column headings in the new table 2. In the example above, simply stating that the RR referred to is the level change would be enormously helpful.
4. The main aim of the paper is to evaluate the impact of stopping elective surgery in winter 2017. In this context the seasonality effects, although interesting, are of secondary interest and should be presented as such. It would be clearer in the results text to focus on the impact of the change on admission numbers, LoS, sex and age etc and leave the seasonality data to a separate section on seasonality. Seasonality is just an annoying thing you are adjusting for rather than of primary interest.
5. Related to the previous point is that there are lots of comparisons being made in the models. There is a danger of inflation of error rates through multiple testing. With this in mind the study should be more clearly flagged as exploratory in nature rather than one that tests a specific hypothesis. The title, abstract and methods section should perhaps state this explicitly.
6. Going back to the methods section and how it sets us up for the results section, I am not clear exactly how the models are being constructed. You mention 'outcomes' in the statistical analysis section for the first time. From the list of data in the section above it is not clear what is an outcome and what is a covariate. From the figures presented and the layout of table T2, I assume that the only thing the model were adjust for is seasonality, but I am not certain. Can I suggest the outcomes and covariates be listed under different headings so that this is clearer?
7. As an additional comment to point 6, is there an argument that some of the models should be adjusted for demographic factors (age, sex, comorbidity etc) rather than these variables being treated as outcomes in their own right. This also relates to point 4 above. The main observations being made regarding sex, age, deprivation and comorbidity relate to seasonality, not the primary research question. Although this is fascinating and not something I would have expected, it seems like a bit of a tangent to what the manuscript is really looking at. I think this could be better framed as secondary analysis (or even a separate paper). The fact that the analysis was an ITS seems irrelevant to these findings and a simple regression model would have shown the same thing.
8. I found the results for elective vs emergency, bed occupancy, private vs public and LoS more interesting than the demographic data and would give them more prominence (2 of the figures are only as supplementary material). These findings are really interesting with regard to the primary research question.

	Minor comments  1. Define abbreviation RR on first mention and tell us how it was derived in the statistical methods section. 2. An additional limitation is the lack of comparison with the wider NHS in England. Were the trends seen genuinely due to stopping elective services or was there a similar wider trend in England? A comparison of activity for the rest of England would be of interesting, although I am not suggesting the authors add these data in. 3. I enjoyed reading the discussion, although it emphasises that issues of seasonality are of secondary interest and that a key focus is on patient numbers, elective capacity, private provision etc.
--	---

VERSION 1 – AUTHOR RESPONSE

Reviewer 1 - Dr. Kunal Kulkarni, University Hospitals of Leicester NHS Trust	
The authors should be commended for performing this large piece of work. I appreciate that analysis of Trust and HES data for multiple years takes much time and effort. The methodology is certainly robust.	We thank the reviewer for these kind comments.
Overall, I unfortunately have two main concerns with the study, namely 1) the "so what" (i.e. how does this actually benefit readers in improving patient care) and 2) generalisability to other populations/regions.	For many years hospital trusts have faced challenges in their planning and capacity for elective surgery, particularly over winter months due to emergency bed pressures, limiting capacity for common elective surgery such as hip and knee replacement. Our aim in this study was to learn from what happened in winter 2017 when elective surgery was cancelled, to understand what was done differently and how the backlog of patients was addressed. This is particularly relevant now following COVID with the current backlog of patients waiting for elective surgery. In this study in one of the largest elective orthopaedic centres, we have observed an overall gradual decline in activity, with fewer operations performed in winter than summer months. There was an increasing trend in patients having more co-morbidity at the time of surgery. Older patients with co-morbidity were more likely to be operated on in summer months. There was an increasing ratio of emergency to elective admissions, in bed occupancy rates, and a step change after winter 2017 with more NHS funded operations being

	done in the private sector who are doing less complex patients. With this being the picture facing elective surgery prior to the impact of COVID. It is important to understand changes that have been happening over time in elective surgery and the impact this is having on services being provided by NHS trusts. The so what and impact on patients is that outsourcing of less complex hip and knee replacements to take advantage of spare capacity in non-NHS hospitals may be a good strategy to reduce waiting lists for surgery, but this leaves the NHS Trust to cope with more complex cases and has training implications because trainee surgeons are usually trained by first undertaking less-complex cases on healthier patients. There are also potential equity implications, if less complex patients have the option of surgery with shorter waiting times at independent providers, whilst more complex (and potentially more deprived) patients do not. We acknowledge that this is the experience of just one trust that is one of the larger elective orthopaedic centres, and hence the findings may not be generalisable to or reflect the experience of other trusts. However, the concept and approach we demonstrate in this study should be informative for other commissioning groups and hospital Trusts to adopt for monitoring of their own elective surgery and capacity.
The authors highlight that they have not included data from the independent sector; however, as they also comment, "the ratio of public to private provision of hip and knee replacements in the local area dropped after winter 2017". Given that the independent sector has played a role in most regions during recovery periods, it would be interesting to know what trends these (perhaps not nationally insignificant) numbers highlight. I appreciate that obtaining this data may not be possible, although as one of the goals is to explore trends, this does represent a data gap,	We have reported on NHS-funded independent provision using the HES-APC data as described in 'Methods – Data Sources' page 6 line 132. This is how we could calculate the public-to-private provision ratio for the area provided in the paper. We have mentioned in the limitations sections (page 3 lines 64-65; page 14 lines 325-326) that our data only allows us to report on NHS-funded independent care, and therefore there could be

particularly given the range of different independent providers in different regions.	a data gap where people are paying privately for their own care via a private provider. Unfortunately, we did not have access to data about privately-funded privately-provided care.
The authors suggest that the merits/uses of these are through "monitoring changes in planning and delivery of elective surgery", although they also state that "the catchment area of the Trust is not exactly the same as the major local CCG and is difficult to define exactly". This raises the question of generalisability, both locally, and subsequently nationally to other regions where policies and strategies may have differed on working through waiting lists.	We believe this statement that these types of trend analysis methods could be useful for other trusts or nationally (page 3 lines 55-58) are still valid. Results/findings may be different in different areas, but applying similar trend analyses could be informative. We accept in our limitations that the catchment area of the Trust will not exactly overlap with the local CCG (page 14 lines 322-323), although do point out that over 89% of admissions at the Trust are from residents of the local CCG, so it's a fairly reasonable overlap. It is a methodological problem without a definite solution to identify the 'catchment area' for a hospital provider. We can identify who actually attended the hospital from the hospital data, but there will be some admissions from areas outside the local area, sometimes people who are resident quite far away. Determining the exact underlying population served by a provider (even when some of that population don't attend the hospital) will always require some assumptions. Here we have taken the large local CCG as the underlying population, provided an explanation of this, and accepted this is not an exact catchment area for the Trust in the limitations with some description of the overlap between CCG and Trust catchment area. We think it is difficult to add much to this, and although it is not a perfect overlap between CCG residents and Trust catchment area, that our analysis of the NHS-funded care provision (public/private) of the CCG residents is a reasonable estimate for the Trust catchment area. We accept that different areas may have different policies/strategies, that we are reported results from one Trust, and these results may not be the same everywhere (limitations sections page 3 lines 62-63; page 14 lines 314-

	316). However, we report results for one of the larger orthopaedic centres (as mentioned on page 14 line 314), and we believe that exploring trends and strategies in this centre may be informative for others.
In their conclusions, the authors highlight 2-3 existing studies that each agree with part of this study's conclusions, raising a query about the novelty of these findings and the unique points they bring to the evidence base. There are also other similar studies that I have read during the past couple of years that highlight similar trends observations peri-COVID.	In the 'comparison to other studies' section (page 14-15 lines 327-339) we have pointed out other studies that suggest an increasing use of independent providers for elective hip/knee surgery, and similar impact of patient characteristics on length of stay for hip/knee replacements. To some extent this is reassuring that certain findings using our dataset are consistent with other studies, suggesting wider generalisability. It is also interesting that the reviewer has read some similar studies of trend observations peri-COVID, and it was always a supposition that some of our findings may be relevant to the COVID period, although it isn't the time period we were exploring in the study. We still believe we are providing a novel exploration of what happened to elective hip/knee surgery during a time of extreme pressure (winter 2017 elective cancellations), and that some of our findings around changes immediately following those cancellations and seasonality are novel and interesting.
With regard to suggestions/strategies/approaches going forward, the authors state "outsourcing of less complex hip and knee replacements to take advantage of spare capacity in non-NHS hospitals may be a good strategy to reduce waiting times and waiting lists for surgery and get the best results for patients given the evident capacity limitations - while I do not disagree with this statement (and indeed this has been policy many regions both, peri-pandemic and during past winter), this is not a statement specifically supported by data from the current study, and I do not believe this can therefore constitute a true 'learning point', particularly when no supplementary material is provided on how exactly patients could be prioritised for this approach (other than being low ASA).	We think that this is something that is observed after the winter 2017 cancellations in our trend analyses as summarised in the 'principal findings' section (page 13 lines 289-308). The public-to-private provision ratio dropped, whilst age and comorbidities of people treated in the hospital increased, suggesting an outsourcing of less complex patients to independent providers. Observing this strategy at a large orthopaedic centre may mean it is applicable elsewhere. However, we agree with the reviewer that from the data in this study we cannot say how this impacts on patient outcomes for different sub-groups of patients, or how it impacts the overall efficiency of elective care at the hospital, and

	therefore it is difficult to recommend who should be outsourced using our study data, except what is being observed at the Trust. We have added a caveat about quality of patient outcomes to the end of the first paragraph in 'implications for clinicians and policy makers' (page 15 lines 341-349). We would also point to the lines in the second paragraph where we summarise that it is important to understand how any strategies might impact throughput of patients, waiting times, waiting lists, outcomes of surgery, costs, and equity of access to surgery (page 15 lines 355-357). We have added a similar caveat to the limitations section (page 14 lines 316-319).
Ultimately, the aim of this study was to "understand what happens after common, planned elective surgery is temporarily cancelled, and how this might inform optimum planning of elective surgery when capacity is limited, such as following the COVID-19 pandemic." However, the discussion unfortunately fails to highlight pragmatic and evidence-based solutions to the changes in capacity/throughput observed. The authors conclude "the Trust has outsourced less complex patients to independent providers, and/or treated them during winter when capacity is most limited. There is a need to explore whether these are strategies that could be used explicitly to maximise the use of limited elective capacity, provide benefit to patients, and value for money for taxpayers" - leaving this initial aim of 'informing optimum planning' unanswered.	We accept this comment that we do not provide definitive answers in terms of optimum planning of elective surgery when capacity is limited. This is an observational study of what happened at one large orthopaedic centre providing hip and knee replacements after cancellation of elective services in winter 2017. The hope was that by exploring what happened at that Trust, this might provide some suggestions about potential strategies that could help. We observed things like outsourcing of less complex patients to independent providers, and also potentially scheduling less complex patients during busy seasons/times. We accept there is more work that needs to be done to explore the impact of these approaches on various important outcomes (at patient-level/hospital-level/commissioner-level) and that not all of those answers are provided within this study.
In summary, while the highlighted observations are interesting, my main concern is that this study in its current form unfortunately fails to clearly address what can be done to resolve the issues highlighted, and therefore I am unclear how readers with similar problems from other regions will find this of interest in tangibly improving patient care.	Please see our earlier response to the reviewer on the novelty of this study. The key observation and impact on patients is that whilst outsourcing of less complex hip and knee replacements to take advantage of spare capacity in non-NHS hospitals may be a good strategy to reduce waiting lists for surgery, this leaves the NHS Trust to cope with more complex cases and has training implications because trainee surgeons are usually trained by first undertaking less-complex cases on healthier patients. There are also potential equity implications, if less complex patients have the option of surgery with shorter waiting times at independent providers, whilst

	more complex (and potentially more deprived) patients do not.
--	---

Reviewer 2 - Dr. William Gray, NHS Improvement	
This is a paper on an important and interesting topic. The findings have direct relevance to the current interest in recovery of elective services following the COVID-19 pandemic and in building future resilience into elective surgery services.	We thank the reviewer for these kind comments.
Major Comments	
The methods and results sections are difficult to follow in places and it is not clear how the data in the text of the results section relate to the figures and tables. As an example (although this is consistent throughout the results section) the first mention of the regression models in the text is 'Numbers of knee replacements dropped by 16% after winter 2017 (RR=0.843, 95% CI: 0.728 to 0.976, p=0.022),'. It takes a good deal of searching to realise that this is referring to the level change in Supplementary material T2 and only at this point was I able to understand what the figures were referring to. The fact that the abbreviation RR is not pre-defined does not help. I have a few suggestions that might help make the results section easier to follow, although you may not feel that all these need to be followed and there is no intention to be prescriptive.	We accept that interrupted time series methods and results are not straightforward to describe/report and we may not have done the best job of this description. We have tried to improve the description of methods/results for the interrupted time series methods, including following most of these suggestions as described in more detail below.
A simple descriptive table (Table 1) of the patient characteristics at the pre and post periods would be helpful for orientation. As this is an online journal, I assume table/figure limits are not a major issue.	We are a little wary of reporting patient characteristics at a single timepoint or averaged over a time period pre/post winter 2017 as this starts to suggest a single timepoint before/after analysis, whilst this study uses trend/ITS analysis to check for an immediate level change and trend change following the cancellations in winter 2017. We think the plots and ITS results report the before/after changes more carefully, but accept that the explanation of the ITS methods and results is not straightforward and clear so have made attempts to improve this as outlined below. We are also aware that BMJ Open's submission guidelines suggest: "with up to five figures and tables. This is flexible, but exceeding this will impact upon the paper's

	'readability'." As such we have tried not to go overboard with the numbers of tables and figures but have added more for clarity as described below.
ITS analysis can be rather dense, and the figures are a vital visual summary, but I feel the pre-trend, level change and post-trend data in Table T2 would be better moved to the main manuscript as a new table 2 as it is central to the results being presented. The seasonality data is fine as supplemental.	We have created a new Table 1 for the main manuscript with the results of the main interrupted time series analyses as recommended. We have left the supplementary tables the same which include the seasonality results.
The results text should link more closely to the data and column headings in the new table 2. In the example above, simply stating that the RR referred to is the level change would be enormously helpful.	We have attempted to link the description of the ITS analysis in the 'statistical analysis' section (page 7-8 lines 170-189), more closely to the reporting of results by using the same terms for both: pre-trend, level change, and trend change. We have also tried to consistently use these terms when reporting the results in the text, rather than using 'RR'.
The main aim of the paper is to evaluate the impact of stopping elective surgery in winter 2017. In this context the seasonality effects, although interesting, are of secondary interest and should be presented as such. It would be clearer in the results text to focus on the impact of the change on admission numbers, LoS, sex and age etc and leave the seasonality date to a separate section on seasonality. Seasonality is just an annoying thing you are adjusting for rather than of primary interest.	We have separated the trend results and the seasonality results. We have provided clarified trend results in Table 1, with more detailed description in the results section (pages 9-11). And we have separated out the seasonality results which are still provided in Supplementary Table T2, but also described in a separate results section (pages 11-12 lines 261-285).
Related to the previous point is that there are lots of comparisons being made in the models. There is a danger of inflation of error rates through multiple testing. With this in mind the study should be more clearly flagged as exploratory in nature rather than one that tests a specific hypothesis. The title, abstract and methods section should perhaps state this explicitly.	We are aware of the potential danger of inflated type 1 error rates and have flagged this in our limitations (page 14 lines 319-320). We have updated the title, abstract, and methods to try to be clear this a descriptive observational study exploring the trends in elective hip/knee replacements and potential changes in trends after winter 2017. Additionally, we have added a few lines in our limitations section to emphasise the same (page 14 lines 316-319).
Going back to the methods section and how it sets us up for the results section, I am not clear exactly how the models are being constructed. You mention 'outcomes' in the statistical analysis section for the first time. From the list of data in the section above it is not clear what is an outcome and what is a covariate. From	We have attempted to provide a better, clearer explanation of the ITS models in the methods section (page 7-8 lines 170-189). We have conducted separate ITS models for each 'outcome', and also stratified by hips and knees. In each case, the covariables included in the regression model are 'pre-trend', 'level change',

the figures presented and the layout of table T2, I assume that the only thing the model were adjust for is seasonality, but I am not certain. Can I suggest the outcomes and covariates be listed under different headings so that this is clearer?	'slope change', and indicator variables for 'spring', 'summer', and 'autumn' compared to winter.
As an additional comment to point 6, is there an argument that some of the models should be adjusted for demographic factors (age, sex, comorbidity etc) rather than these variables being treated as outcomes in their own right. This also relates to point 4 above. The main observations being made regarding sex, age, deprivation and comorbidity relate to seasonality, not the primary research question. Although this is fascinating and not something I would have expected, it seems like a bit of a tangent to what the manuscript is really looking at. I think this could be better framed as secondary analysis (or even a separate paper). The fact that the analysis was an ITS seems irrelevant to these findings and a simple regression model would have shown the same thing.	The ITS analysis is essentially a fairly simple regression model, which tends to be called segmented regression because it allows the regression line to break at the intervention/interruption point and have a level change, trend (slope) change or both. For the most part, it doesn't need to be adjusted for demographics because the time series acts as self-controlled before/after the intervention point (unless large changes are expected in demographics over time). However, we did want to explore what (if any) changes there might be at or after the intervention point of winter 2017 when elective surgery was cancelled for a few months. We didn't just want to adjust for things like age and sex, but were genuinely interested in whether there would be any trends and trend-changes at winter 2017 in these demographics of who was being admitted for elective hip/knee operations. As it happens some of the findings were not trends changes at winter 2017, but seasonality in some of the demographics – we thought this in itself was interesting as it could indicate intentional scheduling of particular patients during easier/harder seasons or times of year.
I found the results for elective vs emergency, bed occupancy, private vs public and LoS more interesting than the demographic data and would give them more prominence (2 of the figures are only as supplementary material). These findings are really interesting with regard to the primary research question.	We found all of our results/figures interesting and would happily include all of them, but we were aware of the BMJ Open guidance mentioned above that ideally we should limit to 5 figures/tables. Although the overall elective to emergency ratio is fascinating, we felt it was one element of the analysis that did not relate specifically to hip/knee replacements and was a proxy for elective capacity, so included it as a supplementary rather than in the main text. Length of stay, whilst reducing overall, didn't indicate any trend changes at winter 2017 so we felt this could go in the supplementary materials. If the editor is willing to include more figures in the article we'd be happy to move them from supplementary into the main paper.
Minor Comments	

Define abbreviation RR on first mention and tell us how it was derived in the statistical methods section	We have updated our explanation of the interrupted time series methods (pages 7-8 lines 170-191) and the reported results to use the terms pre-trend, level change, and trend change, and got rid of 'RR' in the results section.
An additional limitation is the lack of comparison with the wider NHS in England. Were the trends seen genuinely due to stopping elective services or was there a similar wider trend in England? A comparison of activity for the rest of England would be of interesting, although I am not suggesting the authors add these data in	We do include some comparison to national studies in our 'comparison to other studies' section (pages 14-15, lines 327-339) although these studies did not explore the impact of winter 2017 cancellations. The winter 2017 elective cancellation (for hip/knee replacements) was nationwide so we would expect all Trusts to have had to take measures to cope with this: https://www.theguardian.com/society/2017/dec/21/nhs-cancels-surgery-tens-of-thousands-avoid-winter-crisis. However, we accept we have focussed on a local Trust with which we had a collaborative partnership and do not make comparisons with other Trusts. We have mentioned this in our limitations section (page 14 lines 314-316).
I enjoyed reading the discussion, although it emphasises that issues of seasonality are of secondary interest and that a key focus is on patient numbers, elective capacity, private provision etc.	We have separated the trend results and the seasonality results. We have provided clarified trend results in Table 1, with more detailed description in the results section (pages 9-11). And we have separated out the seasonality results which are still provided in Supplementary Table T2, but also described in a separate results section (pages 11-12 lines 261-285).

VERSION 2 – REVIEW

REVIEWER	Kulkarni, Kunal University Hospitals of Leicester NHS Trust, Trauma & Orthopaedics
REVIEW RETURNED	11-Feb-2023

GENERAL COMMENTS	Thank you to the authors for their replies to the initial review. also note the edits made to the manuscript, primarily to the methods and results sections in response to the 2nd reviewer's comments; these do help increase clarity. The previously raised positives (e.g. being well written, clear goal, large dataset, relevant topic) all remain. However, my major concerns with the manuscript - namely clinical relevance, generalisability, and novelty - still remain and (notwithstanding the authors having stated most of these in their
--

limitations), there do not appear to have been many edits to the manuscript to address these.

As previously highlighted, other reports and papers highlighting similar challenges do already exist. For example -

<https://cabinet.leicester.gov.uk/documents/s98190/HWB%20-%20Emergency%20and%20the%20impact%20on%20Planned%20Surgeries.pdf> this document I was able to find online from my own Trust highlights the very issues the authors raise re: the 2017 winter pressures, but also outline a tangible locally adopted solution - one of the major limitations of this observational series.

The authors comment on the challenges/impact of outsourcing routine work to the private sector and the ensuing adverse impact on the NHS. I agree with this point, although again, it has already been made by other studies/reports:

- <https://www.ncbi.nlm.nih.gov/pmc/articles/PMC2249623/>

-

<https://www.sciencedirect.com/science/article/pii/S0047272718301464>

- <https://www.bma.org.uk/media/5378/bma-nhs-outsourcing-report-march-2022.pdf>

-

<https://www.sciencedirect.com/science/article/pii/S2468266722001335>

The impact on education and training from winter pressures are made in one sentence by the authors, but it maybe worth including references to these, for example these statements raised by ASiT and the BOA:

- <https://www.asit.org/resources/archived-articles-documents/2018/winter-pressure-affecting-surgeons-in-training/res1289>

- <https://www.boa.ac.uk/resources/boa-statement-on-training-and-winter-pressure.html>

Additional resources to consider to address the lack of potential solutions for readers, include these RCS reports that attempt to propose 'potential solutions':

- <https://www.rcseng.ac.uk/news-and-events/news/archive/guidance-to-help-manage-winter-pressure/>

- <https://www.rcseng.ac.uk/about-the-rcs/government-relations-and-consultation/position-statements-and-reports/the-case-for-surgical-hubs/>

Overall, not much has been edited to address the concerns I had with the original manuscript, although this does not take away from the fact that the authors have analysed a large dataset to provide objective local data via a well written manuscript that others may wish to consider in developing future solutions (albeit from data that is now >5 years old).

I will defer to the editor to determine whether they feel there is sufficient novelty in the paper and relevance to readers to propose publication; if they feel there is merit then I would suggest either 1) including some of the points raised from the above references to add weight and evidence to support the (currently not wholly substantiated) points raised in the discussion, or 2) significantly reducing the length of the manuscript to make the manuscript/message more punchy and isolate the paper to only the

	objective findings observed and limiting/removing speculative discussion on 'next steps' beyond the remit/support of the current data.
--	--

REVIEWER	Gray, William NHS Improvement, Getting It Right First Time
REVIEW RETURNED	16-Feb-2023

GENERAL COMMENTS	The manuscript is much improved and much easier to read following the changes made. I only have two additional comments:-  1. The limitations should acknowledge the potential for confounding between the outcomes studied. There is an obvious relationship between age and frailty, but a less obvious one between greater age and lower deprivation in ONS data. My guess is that, for knees, the association with deprivation in the time series may be a form of proxy for the relationship noted for age. The authors should acknowledge this point. 2. The authors suggest that the trend to increasing age over time is due to out-sourcing younger, fitter patients to the private sector. This is likely part of the explanation. There is also likely to be an effect from suspending surgery: patient who have to wait longer for surgery will on average be older than those who didn't have to wait as long pre 2017. This should also be acknowledged.
---

VERSION 2 – AUTHOR RESPONSE

Reviewer 1 - Dr. Kunal Kulkarni, University Hospitals of Leicester NHS Trust	
Thank you to the authors for their replies to the initial review. I also note the edits made to the manuscript, primarily to the methods and results sections in response to the 2nd reviewer's comments; these do help increase clarity. The previously raised positives (e.g. being well written, clear goal, large dataset, relevant topic) all remain. However, my major concerns with the manuscript - namely clinical relevance, generalisability, and novelty - still remain and (notwithstanding the authors having stated most of these in their limitations), there do not appear to have been many edits to the manuscript to address these.	Thanks for the advice to improve the manuscript and apologies that we haven't managed to address all of the concerns.
As previously highlighted, other reports and papers highlighting similar challenges do already exist. For example - https://cabinet.leicester.gov.uk/documents/s98190/HWB%20-%20Emergency%20and%20the%20impact%20on%20Planned%20Surgeries.pdf - this document I was able to find online from my own Trust highlights the very issues the authors raise re: the 2017 winter pressures, but also outline a	We have expanded the section on 'implications for clinicians and policy makers' and acknowledge that other trusts have adopted similar solutions including this reference on page 15 line 351.

tangible locally adopted solution - one of the major limitations of this observational series.	
The authors comment on the challenges/impact of outsourcing routine work to the private sector and the ensuing adverse impact on the NHS. I agree with this point, although again, it has already been made by other studies/reports: https://www.ncbi.nlm.nih.gov/pmc/articles/PMC2249623/ https://www.sciencedirect.com/science/article/pii/S0047272718301464 https://www.bma.org.uk/media/5378/bma-nhs-outsourcing-report-march-2022.pdf https://www.sciencedirect.com/science/article/pii/S2468266722001335	We have added more information about NHS-funded outsourcing of services to independent providers, including these references, on pages 15-16 lines 353-357.
The impact on education and training from winter pressures are made in one sentence by the authors, but it maybe worth including references to these, for example these statements raised by ASiT and the BOA: - https://www.asit.org/resources/archived-articles-documents/2018/winter-pressures-affecting-surgeons-in-training/res1289 - https://www.boa.ac.uk/resources/boa-statement-on-training-and-winter-pressures.html	We have added a paragraph about the impact on training and education in the 'implications for clinicians and policy makers' on page 16 lines 359-363.
Additional resources to consider to address the lack of potential solutions for readers, include these RCS reports that attempt to propose 'potential solutions': - https://www.rcseng.ac.uk/news-and-events/news/archive/guidance-to-help-manage-winter-pressures/ - https://www.rcseng.ac.uk/about-the-rcs/government-relations-and-consultation/position-statements-and-reports/the-case-for-surgical-hubs/	We have mentioned the RCS advice around trainees in the paragraph on training and education, page 16 lines 362-363.

Reviewer 2 - Dr. William Gray, NHS Improvement	
The manuscript is much improved and much easier to read following the changes made.	Thanks for the advice on improving the manuscript.
The limitations should acknowledge the potential for confounding between the outcomes studied. There is an obvious relationship between age and frailty, but a less obvious one between greater age and lower deprivation in ONS data. My guess is that, for knees, the association with deprivation in the time series may be a form of proxy for the relationship noted for age. The authors should acknowledge this point.	We have acknowledged this point in the 'strengths and limitations' section, page 14 lines 323-325.
The authors suggest that the trend to increasing age over time is due to out-sourcing younger, fitter patients to the private sector. This is likely part of the explanation. There is also likely to be an effect from suspending surgery: patient who have to wait longer for surgery will on average be older than those who didn't have to wait as long pre 2017. This should also be acknowledged.	We have acknowledged this point in the 'strengths and limitations' section, page 14 lines 325-326.